# A Simple Method for a Protective Coating on Stainless Steel against Molten Aluminum Alloy Comprising Polymer-Derived Ceramics, Oxides and Refractory Ceramics

**DOI:** 10.3390/ma14061519

**Published:** 2021-03-19

**Authors:** Sébastien Quenard, Marilyne Roumanie

**Affiliations:** CEA, LITEN, Université Grenoble Alpes, 17 Rue des Martyrs, F-38054 Grenoble, France; marilyne.roumanie@cea.fr

**Keywords:** coating, polymer-derived ceramics, high temperature, corrosion barrier, liquid aluminum alloy

## Abstract

A new coating based on polymer-derived ceramics (PDC), oxides and refractory ceramic with a thickness of around 50 µm has been developed to improve the resistance corrosion of stainless steel substrate against molten aluminum alloy in a thermal energy storage (TES) system designed to run at high temperature (up to 600 °C). These coatings implemented by straightforward methods, like tape casting or paintbrush, were coated on planar and cylindrical stainless-steel substrates, pyrolyzed at 700 °C before being plunged for 600 and 1200 h in molten AlSi_12_ at 700 °C. The stainless-steel substrate appears healthy without intermetallic compounds, characteristic of molten aluminum alloy corrosion. The protective coating against AlSi_12_ corrosion shows excellent performance and appears interesting for TES applications.

## 1. Introduction

Renewable energies such as solar energy are intermittent resources that can induce a mismatch between supply and demand and constitute a limit to their use. Thermal energy storage (TES) is essential in increasing the supply and use of renewable energy and reducing the carbon footprint. The integration and utilization of latent thermal energy storage (LTES) with heat recovery systems is the most potential and cost-effective solution. Due to the large energy storage density of metallic phase change materials (PCM), combining solar power plants with LTES is the most effective method to provide flexible electricity to the grid and supply large-scale power services [1].

Many PCMs have been reported in the literature for concentrating solar power (CSP) such as Mg-51% Zn [2], light-weight alloys based on Mg-Zn-Al [3], CaSi (melting temperature (Tm) of 782 °C) [4], CuMgSi (Tm of 742 °C) [4], eutectic alloy compositions based on 88Al-12Si (AlSi_12_–Tm of 577 °C) [5], and 60Al-34-Mg-6Zn (Tm of 454 °C) [6].

AlSi_12_ alloy is considered to be a promising metallic PCM [7]. It has a low melting temperature [8], is stable during heating and cooling cycles [9], has high thermal conductivity (190 W·m^−1^·K^−1^ at 577 °C [7]), high latent heat of fusion of 548.6 J·g^−1^ at 577 °C [9] and is low cost.

Liquid aluminum alloys are materials known to be extremely corrosive to most metals and metal oxides [10], which is critical in a TES system designed in stainless steel and working at a temperature above 600 °C. Corrosion issues in TES systems conditions are hardly addressed in the literature [11], although the literature is well documented concerning the development of corrosion-resistant materials. Among the investigations, boronized carbon steel showed good resistance to corrosion [12] following 120 h at 630 °C in molten aluminum. This working time is, however, limited compared to that of the TES systems.

Fukahori et al. [13] introduce the high corrosion resistance of ceramic substrates such as alumina (Al_2_O_3_), aluminum nitride (AlN), and silicon nitride (Si_3_N_4_) to molten Al-Si alloys. However, these interesting materials for the LTES are brittle, expensive, and difficult to deposit with low-cost processes. To make these materials suitable for the application, a coating of iron container surface with a protective layer of Al_2_O_3_ has been proposed [11]. The method consists of hot dipping the steel into molten aluminum. Molten aluminum in contact with iron creates iron aluminide (intermetallic compound). The surface is then oxidized to obtain the protective ceramic layer [11]. Although the Al_2_O_3_ layer shows excellent corrosion resistance at 250 °C for 250 h, no information is given for applications working above 600 °C. The sol-gel process enables the synthesis of ceramic material as silica (SiO_2_), zirconia (ZrO_2_), Al_2_O_3_, titanium oxide (TiO_2_). These materials, easily deposited on surfaces with inexpensive processes, showed excellent chemical stability, and improve the corrosion resistance of metal substrates (steel, aluminum, and their alloys) at low temperatures [14].

Polymer-derived ceramics (PDC) [15] are organic/inorganic polymers, forming amorphous ceramics after pyrolysis at temperatures between 600 and 1000 °C. PDC are easy to apply on substrates of any shape by dip coating, spray-coating, spin-coating, or tape casting. PDC composite coatings are described in literature as promising candidates to be used as alternative environmental barrier coatings for corrosion and oxidation protection of metals at elevated temperatures (600–1000 °C) [16]. The main drawback of PDC technology is the unavoidable shrinkage which occurs due to the large density change when the polymer precursor with a typical density of 1 g·cm^−3^, is converted to the ceramic product, often with a density above 2 g·cm^−3^ [17]. The volume shrinkage can be greater than 50%. Active or passive fillers are, therefore, added to limit shrinkage. In the literature based on oxygen corrosion, the addition of glass in the PDC slurry can be noticed [17,18,19]. This glass improves the adhesion of the PDC-containing layer to the substrate and thus improves the corrosion protection of the metal. Al_2_O_3_ [17] or Al_2_O_3_-Y_2_O_3_-ZrO_2_ [18,19] passive fillers are also blended to minimize the shrinkage and for its good antioxidation properties at high temperature. These, although interesting, composite PDCs have not been evaluated in the presence of molten aluminum.

Commercial corrosion protection solutions have also been developed for the aluminum foundry industry (Condat, Dycote, Aluminium Martigny, Chimilin, France). These materials are based on graphite, boron nitride, and refractory powders. However, these solutions have not been listed in the TES literature. A preliminary study based on commercial materials was, therefore, carried out and detailed in the Appendix A. These materials are easy to process and inexpensive. An investigation at 700 °C in molten AlSi_12_ of these materials showed poor adhesion, delamination, or lack of corrosion resistance of the coating on the stainless-steel TES substrate. However, this study confirmed the interest in using the boron nitride material to fight against corrosion of aluminum alloys.

The primary object of this study is to develop a coating on 304L stainless steel with a high corrosion protection in presence of molten AlSi_12_ at 700 °C for 600–1200 h. Besides, to be easy to implement and economically attractive, the relevant coating needs to have the following properties: (i) good adhesion at high temperature, delamination- and crack-free; (ii) low wettability to molten aluminum alloy (AlSi_12_); (iii) good durability in molten aluminum alloy (AlSi_12_).

The composition of the protective layer is based on polymer-derived ceramic, glass frit and passive filler with low interaction with molten aluminum. The thermal behavior, wettability and thermal conductivity of the protective material were first characterized. This material was then applied to 304L stainless steel substrates, and its behavior in the presence of molten AlSi_12_ was evaluated.

## 2. Materials and Methods

### 2.1. Protective Material

Polysilsesquioxane Silres MK (23 wt% Wacker AG, Munich, Germany), a polymer-derived ceramic (PDC) in the solid-state used as a binder in the formulation, was dissolved in a solvent (43 wt% Diestone^®^ Socomore, Vannes, France). The mixture was blended with a glass frit (28 wt%) based on zinc oxide (major elements) and a boron nitride powder having a mean particle size of 0.5 µm (6 wt% Momentive Grade AC6111, NY, USA). The PDC composite slurry was homogenized for 1 h by rotation in a closed container. This PDC has a melting temperature of around 46 °C, which facilitates its shaping. Finally, this preceramic polymer crosslinks from 200 °C and up to 300 °C to become an infusible material. The ZnO-based glass frit was chosen for its high ZnO content (>30 wt%—ZnO is known to have a good elasticity behavior) and for having a low glass transition temperature (Tg = 475 °C). Preliminary tests with PDC and glass frit layer were carried out and are described in Appendix A. Despite a good adhesion, this corrosion barrier was too reactive with molten AlSi_12_ and was not effective (total loss of the initial layer). To overcome the issue, boron nitride filler was blended to the PDC and the glass frit. BN is known for its high thermal conductivity and for low wettability to the molten metal.

### 2.2. Methods

#### 2.2.1. Protective Material Characterization

The PDC composite slurry was spread on a non-wetting polyethylene terephthalate (Mylar) and dried at room temperature. Dried pieces of PDC composite material were then ground using a planetary ball mill (Pulverisette 6, FritschGmbH, Idar-Oberstein, Germany), for 10 min with a fixed speed rate of 200 rpm. The jar and the grinding media were made of TZ3Y (polycrystalline tetragonal zirconia stabilized with 3 mol% of Y_2_O_3_). Grinding media were 2 mm diameter balls and the volume of the jar was 500 mL. PDC composite powder was then sieved at 200 µm. The resulting powder was shaped by thermo-pressing at a temperature of 205 °C and a pressure of 300 bar. The pellets were then pyrolyzed in argon at 700 °C for 1 h. The coefficient of thermal expansion (CTE) and the thermal conductivity of the protective material were determined from these pellets.

The CTE, the glass transition temperature, and the softening point of the PDC composite were performed by dilatometry (SETARAM Kep technologies, Caluire, France —SETSYS thermomechanical analyzer) on the temperature range from 20 to 700 °C with a rate of 3 °C·min^−1^. Two CTE measurements were done, the first time after one thermal cycle up to 700 °C and the second time after five thermal cycles up to 700 °C on a pellet (diameter of 8 mm/height of 7 mm). The CTE was calculated from the following Equation (1) between 20 and 500 °C:(1)$$\alpha =\frac{1}{L_{T0}}\left(\frac{L_{T}-L_{T0}}{T-T_{0}}\right)$$
where α (10^−6^·K^−1^) is the coefficient of thermal expansion, $T_{0}$ and T (K) the initial temperature and the final temperature in the selected temperature range, and $L_{T0}$ and $L_{T}$ (mm), respectively, thicknesses at $T_{0}$ and at T.

Thermal conductivity was measured by the Hot Disk Transient Plan Source (TPS) method according to ISO 22007-2 by the company Thermoconcept [20]. This method described in detail by He et al. [21] is a nondestructive, direct, and fast technique. The Hot disk sensor is placed between two PDC composite pellets (diameter of 25 mm, height of 8 mm). This sensor serves both as a heat source and a temperature sensor. Specifically, the variation of temperature is determined from the change in resistance via a Wheatstone bridge (2):(2)$$R\left(t\right)=R_{0}\left[1+\alpha \Delta T\left(t\right)\right]$$
where R is the total electrical resistance at time t, $R_{0}$ the initial resistance at t=0, α the temperature coefficient resistivity of the nickel (material of the sensor), and the ΔT change in temperature between t=0 and t.

The average temperature increases in the sensor surface following Equation (3):(3)$$\Delta T\left(\tau \right)=\frac{P_{0}}{\pi^{\frac{3}{2}}a\lambda }D\left(\tau \right)$$
where $P_{0}$ is the power output of the sensor, a the radius of the largest ring, λ the thermal conductivity, and $D\left(\tau \right)$ is a function proportional to the temperature rise depending on a dimensionless parameter $\tau =\sqrt{\kappa t}a$ with κ the thermal diffusivity.

The thermal conductivity of protective material was determined at room temperature, 300 °C and 600 °C in air.

#### 2.2.2. Protective Coating

##### Preparation and Tests in Molten AlSi_12_

The slurry was spread on a 304L stainless steel substrate, a material used for storage applications, to assess the protective coating behavior in operation. The stainless-steel samples had a surface roughness of 3.3 µm. The metallic substrates were previously degreased with acetone and immersed in an ultrasonic ethanol bath. The protective layer, in the first step, was deposited by a doctor blade on planar 304L substrate. The layer was crosslinked at 200 °C and pyrolyzed for 1 h at 700 °C in argon. Stainless-steel planar substrates coated with the protective layer were inserted in an alumina crucible containing AlSi_12_ metal to assess the corrosion behavior (Figure 1). The crucible was sealed with high-temperature ceramic glue and then placed in a furnace. The temperature of the furnace was set at 700 °C and the heating rate at 5 °C per minute. At this temperature, the AlSi_12_ is molten. The contact of the liquid aluminum with the sample was performed for 600 and 1200 h.

In the second step, the protective layer was applied by paintbrush on cylindrical components, representative of the application (diameter of 54 mm, height of 200 mm). The layer inside the tube was homogenized by rotation using a roller system. The crosslinking at 200 °C in air and the pyrolysis at 700 °C in argon of the layer was performed on these different parts. This process, inexpensive and straightforward, should be easily implemented into a storage system. The tube was filled with AlSi_12_ alloy, closed with the lids, and treated for 600 h at 700 °C in a furnace (Figure 2).

##### Characterizations

The microstructure of the protective layer after coating on the 304L substrate and after static immersion in AlSi_12_ during 600 and 1200 h was investigated by scanning electron microscopy (Philipps XL30 SEM, Eindhoven, The Netherlands). In parallel to SEM observations, energy dispersive X-ray spectroscopy (EDS, Esprit, Bruker Nano GmbH, Berlin, Germany) analyses in map mode were completed on samples in contact with AlSi_12_ molten alloy. This was especially retained to investigate the aluminum distribution. The surface topography and roughness, evaluated as a root-mean-square (RMS) surface roughness, were studied by microscopy confocal (infiniteFocus—magnification X 50, Bruker Alicona, Graz, Austria).

The adhesion of the protective layer on the 304L substrate was performed with automatic adhesion tests [22,23] related to ASTM 4541. These tests were carried out with the Elcometer 510 automatic pull-off adhesion gauge (La Chapelle Saint Mesmin, France). Aluminum dollies (diameter of 20 mm) were glued with the Araldite 2011 adhesive. Samples were kept at ambient temperature for 24 h before tests. Dollies were pulled at a speed of 0.2 MPa·s^−1^ in the normal direction to the coating surface. The adhesion force was measured once the dollie was no longer in contact with the substrate (Figure A2 in Appendix A). Three modes of fracture result from the characterization: (i) cohesive break (interfacial fracture), (ii) adhesive break (a break between the substrate and the layer), and (iii) glue break (coating adhesion higher than glue adhesion). Tests were carried out using three replicates to ensure the repeatability of the measurement.

The surface free energies of the protective layer crosslinked at 200 °C and pyrolyzed at 700 °C were determined by measuring the contact angle. This measurement allows the determination of the solid/liquid interactions, essential parameters in TES application where the AlSi_12_ in liquid state interacts with the protective layer. The contact angles were measured by depositing 3 µL of a drop of liquid on the layer surface through a calibrated microsyringe and a programmable pump system. The polar and dispersive contributions to the surface energy were obtained through three test liquids (water, diiodomethane, ethylene glycol). Recorded images were analyzed to assess the contact angle. The surface free energies were calculated using the Owens and Wendt, Rabel, and Kaelble (OWRK) method [24]. This method assumes that the SFE ($\gamma_{S}$) is a sum of a polar component ($\gamma_{S}^{P}$) and a dispersive one ($\gamma_{S}^{d}$) (4):(4)$$\gamma_{S}=\gamma_{S}^{d}+\gamma_{S}^{P}$$

The probe liquids used (water, diiodomethane, and ethylene glycol) cover a wide range of properties from very polar water to very dispersive diiodomethane. Table 1 reports the characteristics of the three liquids [25].

## 3. Results and Discussion

### 3.1. Protective Material Properties

The thermomechanical behavior of the material containing polysiloxane, glass frit, and boron nitride was determined by dilatometry (Figure 3). As we can see, the thermomechanical behavior appears similar between 20 and 500 °C after several cycles up to 700 °C. For each measurement, a material expansion of 25 µm was observed in this temperature range, and the calculated CTE is to 6.4 K^−1^ (Figure 3a) and 7.3 K^−1^ (Figure 3b). In Figure 3a, a slope change was observed at 200 °C. This temperature corresponds to the crosslinking temperature of the PDC. The samples being thick (7 mm), a first thermal cycling for 1 h at 700 °C before the study does not seem sufficient to have a complete crosslinking of the PDC. However, this time is sufficient to crosslink thin layers used in the TES application. This expansion can be related to the out-gassing and modification of the PDC. Above this temperature, the behavior of the protective material changes regarding the cycles. A significant shrinkage is observed up to 700 °C after the first thermal cycling. This shrinkage can result from two phenomena. The first one can be related to the presence of the glass frit that has a transition temperature from the vitreous to the liquid state close to 500 °C. The second one concerns the polymer-derived ceramic (PDC). The PDCs show a substantial shrinkage when heated to temperatures above 600 °C, as the organic side chains evaporate, and a porous silicon-oxy-carbide glass is formed [26].

After five cyclings in temperature (Figure 3b), a glass transition temperature (T_g_) is also observed at 500 °C. Above T_g_, glasses become soft and capable of deformation without fracture. The softening temperature is observed at 560 °C. Having a part of the material molten in temperature and solidified by cooling can help absorb thermomechanical stresses and limit layer cracking risk during future storage applications. After five thermal cyclings up to 700 °C, the material seems to be stabilized with a glass behavior.

For temperature energy storage applications, it is interesting that the materials used are thermally conductive to promote exchanges and limit thermal barriers. AlSi_12_ has a thermal conductivity of 190 W·m^−1^·K^−1^ at 577 °C and 304L stainless steel used for TES applications of 16.2 W·m^−1^·K^−1^. The thermal conductivity of the protective material was measured between 0.5 and 0.8 W·m^−1^·K^−1^ for a temperature range between 20 and 600 °C (Figure 4). These experimental data could be interesting for future storage system modeling, even if the thin layer would probably have little impact on global thermal exchange. As a remark, the thermal properties of the commercial protective coating materials listed in Appendix A are not provided.

### 3.2. Protective Coating on Planar 304L Samples

#### 3.2.1. Topology and Adhesion

Figure 5 shows a cross-section of a stainless-steel substrate coated with the protective layer pyrolyzed one time at 700 °C. A crack-free layer appears homogeneously distributed on the surface of the substrate on the different areas observed. This layer presents pores that appear trapped in the PDC composite. These pores are probably due to the off-gassing associated with the beginning of the ceramization of the PDC into silicon oxycarbide. The glass frit has a glass transition temperature of about 500 °C. Therefore, the heat treatment at 700 °C led to a liquid glass which, during cooling, froze and trapped the gas bubbles.

The layer surface appears highly irregular, even suggesting that the layer is cracked (Figure 6a). A confocal microscopic topography of the sample surface (Figure 6b) confirms that the deposited layer is strongly wavy. The root mean squares (RMS) surface roughness, sensitive to large peaks and valleys, is 5 µm confirming the previous observations. The protective layer appears continuous. Considering that the thickness measured on the SEM image is a mean point of the layer. The thickness of the layer is therefore about between 30 and 50 µm.

This continuous protective layer over the entire 304L planar substrate has an adhesion greater than 3 MPa. This value is in line with the adhesion values of the commercial coatings listed in Appendix A. The fracture took place in the glue between the dollie and the layer (glue break), proving that the coating adhesion is strong on the substrate and in the layer (coating adhesion higher than glue adhesion).

#### 3.2.2. Surface Energy

Measurements of contact angles of specific liquids are used to determine the surface free energy (SFE) of the protective layer. The contact angles of distilled water, diiodomethane, and ethylene glycol measured on the protective layer are displayed in Table 2.

The contact angles were measured below 90° using distilled water as the probe liquid. The layer has, therefore, a partial hydrophilic behavior. The hydrophilic character appears to increase after the pyrolysis of the layer. The surface free energies are 22.6 and 26.4 mJ·m^−2^, respectively, for a crosslinked layer and a crosslinked/pyrolyzed layer. This protective layer has low surface energy close to plastics as polyvinyl fluoride (PVF = 30.3 mJ·m^−2^) and polyethylene (PE = 32.4 mJ·m^−2^) and lower than that of stainless steel, 40–60 mJ·m^−2^ [27,28].

It can be noticed that with temperature, the dispersive and polar components are reversed. Thus, stronger interactions appear when the PDC is in a ceramic state since the $\gamma_{S}^{P}$ increases at the expense of the dispersive component. The chemical nature of the PDC evolves between processing temperatures. The PDC in the crosslinked state contains organic bonds that break during pyrolysis. At this stage, there are essentially weak Van der Waals-type interactions due to the higher dispersive component. At 700 °C, the PDC is being ceramized, organic groups split off, and the hydrogen bonds are likely accessible in the matrix. These hydrogen bonds evacuate as hydrogen up to 1000 °C. The presence of these bonds leads to a higher wettability with water and polar liquid [29]. The dispersive component after pyrolysis of the protective layer is low, the wettability to nonpolar material is therefore low.

The low surface free energy combined with a low dispersion component should favor a nonadhesion of metallic liquid in TES application.

### 3.3. TES Application

#### 3.3.1. Planar 304L Substrate

Figure 7 and Figure 8 show cross-sectional interfacial morphologies and EDS analyses of the samples after molten aluminum corrosion, respectively, 600 and 1200 h. The interface exhibited a sandwich structure after 600 h, consisting of a 304L substrate with a protective layer and above aluminum. After 1200 h, the configuration is similar. A gap between the aluminum and the protective layer is, however, observed. This gap is due to the nonadhesion of aluminum after removing the substrate from the AlSi_12_. The protective layer appears continuous over the entire substrate without any cracks after treatment. The pores or gas bubbles previously observed no longer appear to be present in the layer. The glass frit is liquid at a temperature of 700 °C. The pores can, therefore, be filled, and gas bubbles evacuated. The protective layer is intact and does not appear to have been affected by the presence of the molten AlSi_12_.

The 304L substrate appears healthy, suggesting the absence of the Fe-Al intermetallics, characteristic of corrosion. The elemental analyses carried out by EDS show a protective layer represented by the Si, O, and C elements, 304L substrate by Fe, and the aluminum alloy consisting of Al and Si. The Al element is only observed above the surface of the protective layer. These observations performed after 600 and 1200 h confirm the protective aspect of this layer against molten AlSi_12_.

#### 3.3.2. Components for TES Application

Figure 9 shows the tube after aging in AlSi_12_ during 600 h at 700 °C and after taking a sample to carry out characterizations.

Figure 10 displays a cross-sectional morphology and EDS analysis of the stainless-steel tube coated with the protective layer after liquid aluminum aging during 600 h. After removing the samples from the representative TES component, the absence of adhesion between AlSi_12_ alloy and container was noticed. An aluminum-free area above the protective layer filled by the SEM resin is, therefore, observed on the SEM image. The thickness of this layer is between 20 and 50 µm. The protective layer has also non-open cavities on the stainless-steel substrate. The cavities have elongated shapes with rough surfaces or boot-like shapes. AlSi_12_ does not seem to penetrate into the layer, otherwise solid aluminum would have torn off the layer when the sample was taken. The presence of cavities is probably due to the coating process combined with higher thermomechanical stresses in cylindrical substrate compared to planar substrate. The thickness variation of the layer does not seem to affect the effectiveness of this protective layer since the 304L substrate is healthy without any corrosion points. EDS analysis did not detect any aluminum in the layer or in the substrate. The protective layer can, therefore, be considered an excellent barrier against AlSi_12_ corrosion and looks promising for TES applications.

## 4. Conclusions

A comparative study of different anticorrosive barriers for 304L stainless steel against molten AlSi_12_ showed low performance over long periods. A composite based on polymer-derived ceramics, ZnO-based glass frit and boron nitride powder was, therefore, developed and applied to stainless steel surfaces by tape casting and brushing. Characterizations of wettability and adhesion, aging tests of the coating in contact with an aluminum alloy (AlSi_12_) at 700 °C for 600 and 1200 h, show high performance to the already studied and existing solutions with a significant increase in durability. This solution also makes it possible to envisage low-cost technical processes for future applications such as energy storage based on metallic phase change materials and applications in aluminum alloy foundry. This solution should be further optimized in the future, especially for the industrial application process of the protective layer (spraying, etc.).

## Figures and Tables

**Figure 1 materials-14-01519-f001:**
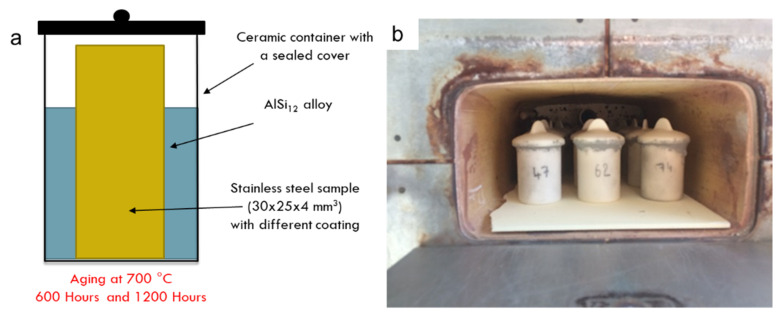
Corrosion resistance of a 304L stainless steel coated with a protective layer in molten AlSi_12_ during 600 and 1200 h at 700 °C (**a**) schematic view and (**b**) ceramic containers in the furnace.

**Figure 2 materials-14-01519-f002:**
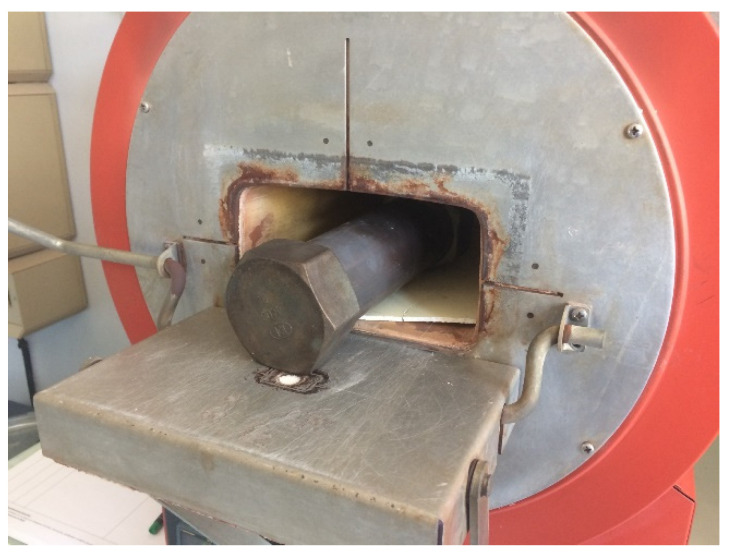
Small representative component coated with protective layer and filled with molten AlSi_12_.

**Figure 3 materials-14-01519-f003:**
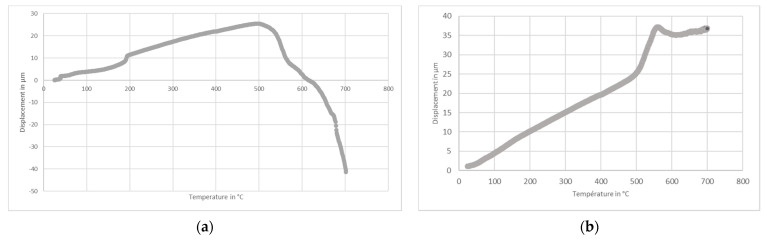
Displacement measurement vs. temperature on SETSYS thermomechanical analyzer (**a**) after one cycling in temperature up to 700 °C and (**b**) after five cyclings in temperature up to 700 °C.

**Figure 4 materials-14-01519-f004:**
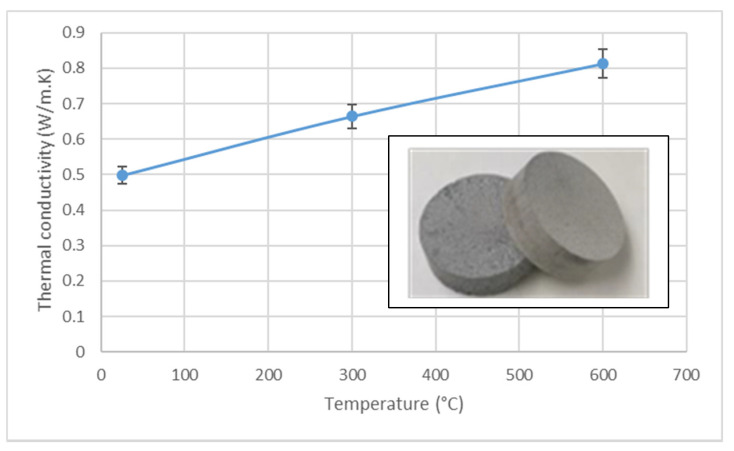
Thermal conductivity of the protective material at 20, 300, and 600 °C.

**Figure 5 materials-14-01519-f005:**
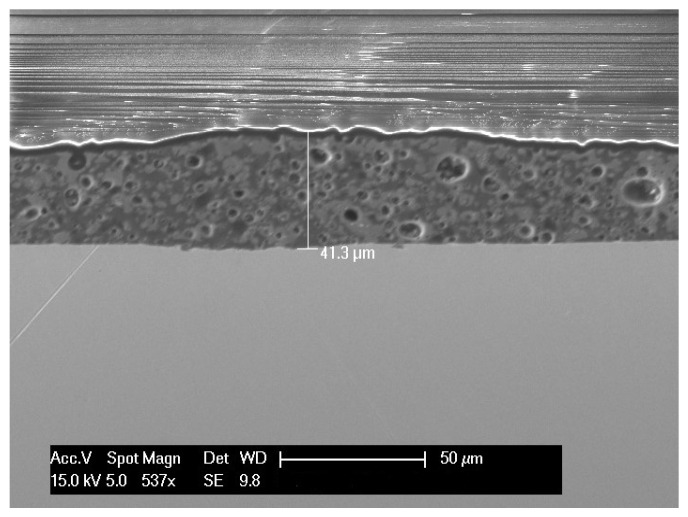
A cross-sectional SEM image of the protective layer coated on 304L stainless steel substrate, crosslinked at 200 °C and pyrolyzed at 700 °C in argon.

**Figure 6 materials-14-01519-f006:**
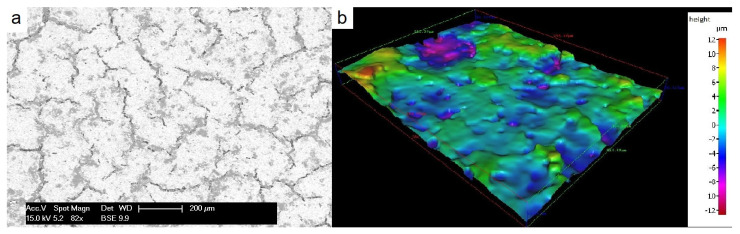
(**a**) SEM image of the surface of protective layer coated on 304L stainless steel substrate, crosslinked at 200 °C and pyrolyzed at 700 °C in argon; (**b**) 3D topography measurement of the layer.

**Figure 7 materials-14-01519-f007:**
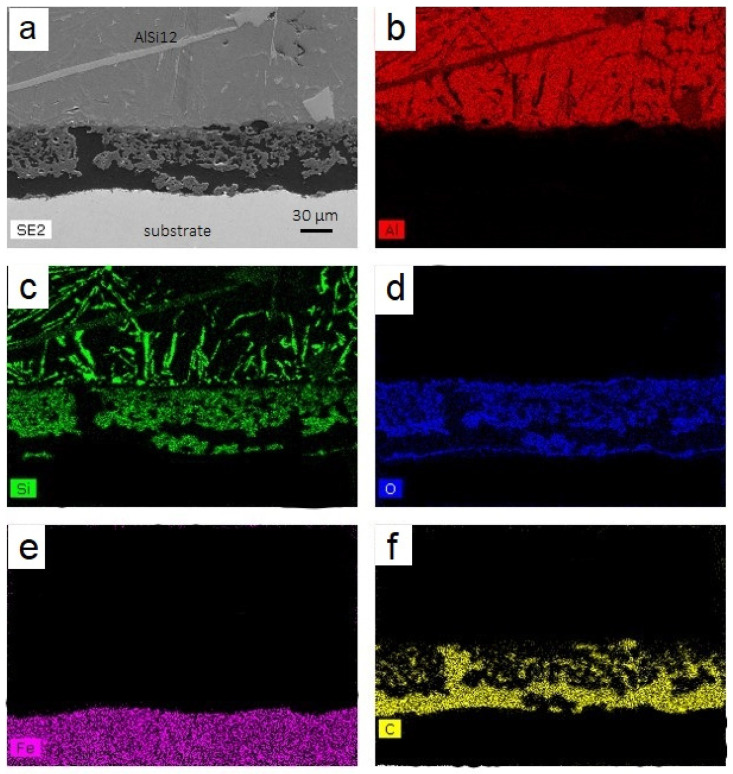
(**a**) Cross-sectional SEM observation of the protective layer coated on the 304L substrate after 600 h at 700 °C in AlSi_12_ (300× magnification) and EDS analysis of Al (**b**), Si (**c**), O (**d**), Fe (**e**), and C (**f**).

**Figure 8 materials-14-01519-f008:**
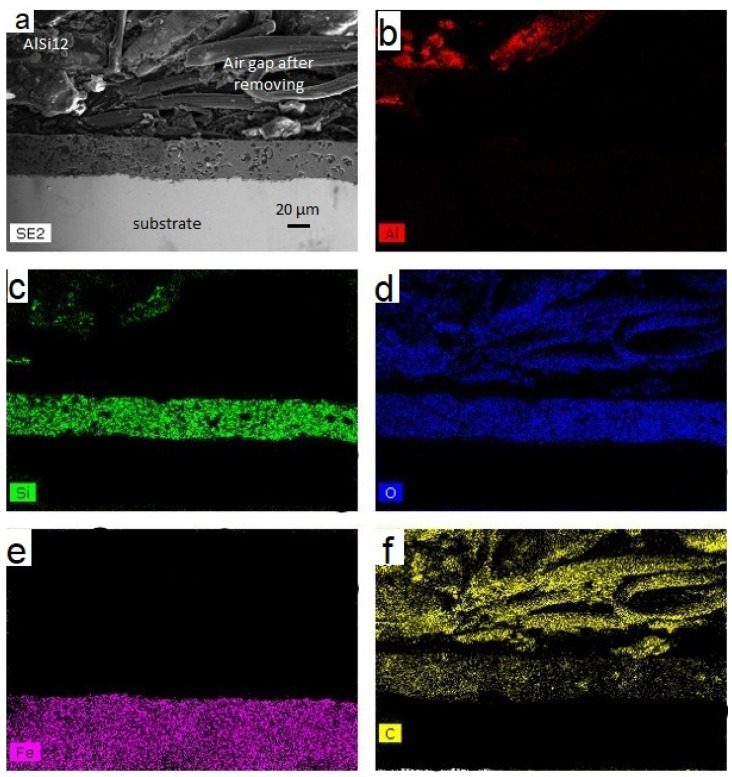
(**a**) Cross-sectional SEM observation of the protective layer coated on the 304L substrate after 1200 h at 700 °C in AlSi_12_ (300× magnification) and EDS analysis of Al (**b**), Si (**c**), O (**d**), Fe (**e**), and C (**f**).

**Figure 9 materials-14-01519-f009:**
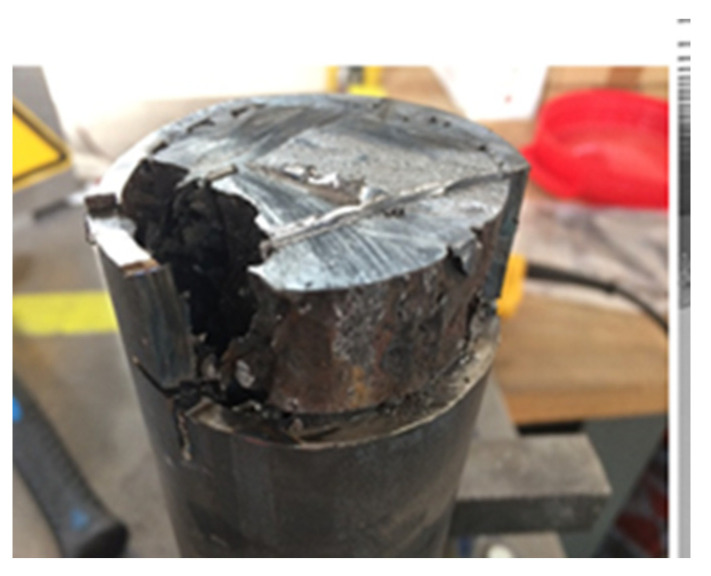
Small representative component coated with protective layer after test in AlSi_12_.

**Figure 10 materials-14-01519-f010:**
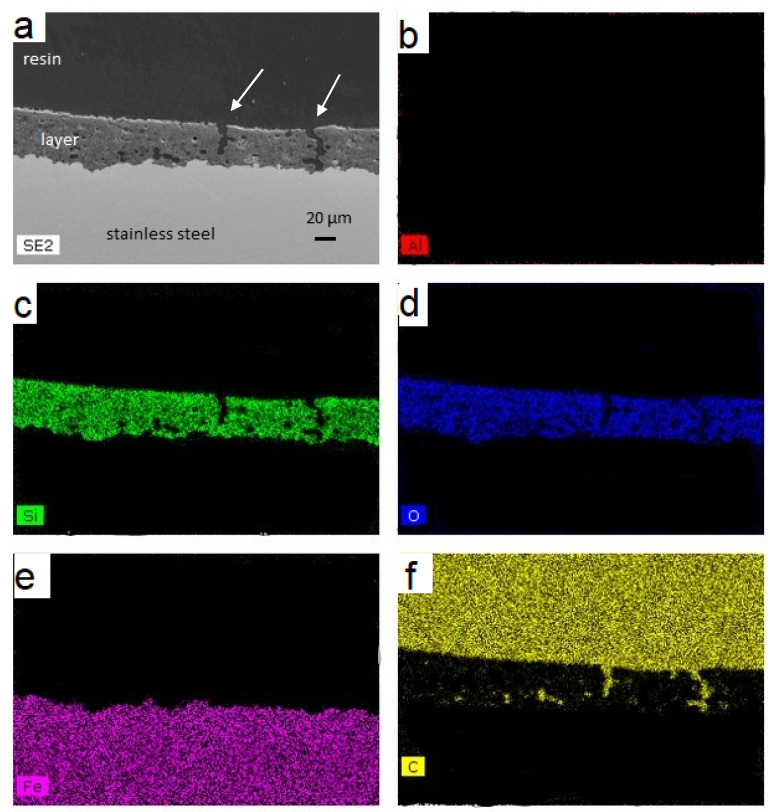
(**a**) SEM image of the protective layer on the 304L tube after aging during 600 h at 700 °C in liquid AlSi_12_ (300× magnification) and EDS analysis of Al (**b**), Si (**c**), O (d), Fe (**e**), and C (**f**).

**Table 1 materials-14-01519-t001:** Dispersion and polar components and surface free energy of water, ethylene glycol and diiodomethane.

Liquid	$\mathrm{Dispersion}\;\mathrm{Component}\;\gamma_{S}^{d}$	$\mathrm{Polar}\;\mathrm{Component}\;\gamma_{S}^{P}$	$\mathrm{Surface}\;\mathrm{Free}\;\mathrm{Energy}\;\gamma_{S}(\mathrm{mJ}\cdot m^{-2})$
Water	21.8	51	72.8
Ethylene glycol	29	19	46.1
Diiodomethane	46.6	4.2	50.8

**Table 2 materials-14-01519-t002:** Surface free energy $\gamma_{S}$ and its components of the protective layer following different surface treatments.

Surface Treatment	Contact Angle Ɵ (°)			Surface Free Energy and Its Components (mJ/m^2^)		
	Water	Diiodomethane	Ethylene Glycol	$\gamma_{S}$	$\gamma_{S}^{d}$	$\gamma_{S}^{P}$
Crosslinking at 200 °C in air	88.1	72.7	82.0	22.6	19.2	3.4
Pyrolysis at 700 °C in Ar	69.4	89	82.6	26.4	9.8	16.6

## Data Availability

Data sharing is not applicable for this article.

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
