# Peer review of "A Simple Method for a Protective Coating on Stainless Steel against Molten Aluminum Alloy Comprising Polymer-Derived Ceramics, Oxides and Refractory Ceramics"

_materials, 2021, doi:10.3390/ma14061519_

Round 1

Reviewer 1 Report

The publication is about a polysilsesquioxane-based coating with glass and boron nitride fillers. It has been developed to improve the resistance  of stainless steel against aluminum melts. A specific aluminum alloy melt is a metallic phase change material (mPCM) used in latent thermal energy storage (LTES).

In the present form the manuscript seems not to be appropriate for publishing in "Materials". It contains too much formatting and spelling errors. The EDX images are not meaningful. In addition, the image quality is insufficient, from which no statement can be made about the layer composition. The influence of the existing open porosity on the coating stability was not investigated. The references to already existing glass-filled PDC coating systems is completely missing. There are some more inconsistencies.

As the scientific topic is interesting for a broader readership, I recommend to revise the manuscript comprehensively before a resubmission. The authors should also improve the english language.

Reviewer 2 Report

The topic of this research is very interesting and significant from the practical side. The investigations in this manuscript are new and original.

The research was extensive, well planned and conducted, and the explanations in the text were understandable and logical. The authors have clearly demonstrated the ability of use composite based on Polymer Derived Ceramics, glass frit based on ZnO, as protective coating on stainless steel exposed to the aggressive molten aluminium alloy AlSi12 at 700 oC.

My suggestion is a small change in the title of the article (adding the word “stainless steel” and “molten” in front of words aluminum alloy, so the proposed title will be: A simple method for a protective coating on stainless steel against molten aluminum alloy comprising Polymer-Derived Ceramics, oxides and refractory ceramics

There are some minor typewriter mistakes in the text which I have marked in the pdf. version of manuscript.

  • In line 52 word instead of wordspostponed one should write proposed
  • In the line 77 please write an abbreviation next to the word boron nitride powder, because which abbreviated form BN was introduced in the line 83.
  • In the line 128 infinite focus Alona equipment – the equipment is InfiniteFocus Alicona
  • In the line 262 please add one space between and Figure 7

Also, I found lots of errors in reference section which I have also marked. Authors must use only one  citation style, but they mixed different citation styles.  Errors in reference section led to the appearance of the “Error! Reference source not found” in the text. In some references, the authors wrote the initials of names with surnames without spaces between them (ref. [2] P.blanco-Rodriguez; ref. [3] S.Doppiu; ref. [5] S.Khare, C.Knight…). Also, some in some references, surnames on the authors are in the first place followed by initial of the first names (e.g. ref. [10] Deqing W; ref. [18] Owens D.K., Wendt R.C. …). Sometimes initials were separated from surnames with comma. Please correct and uniform the writing of references.

Reviewer 3 Report

The article is interesting and has a good visual design  I think that the results of this work deserve publication in Materials journal, because the scientific quality of the paper is good. Some of the improvements I'd recommend are listed in the attached file.

Round 2

Reviewer 1 Report

The spelling and formatting errors have been corrected. References to glass-filled PDC-systems have been added in the introduction. According to the authors, the influence of porosity will be investigated in a future project.

The comments were implemented carefully.

However, in Fig. 7, 8 and 10, the SEM images on the left side do not match with the shown EDX images. The section where the EDX image was taken should be marked in the SEM image on the left, because the image quality of the SEM image shown at the EDX images is not sufficient.

Author Response

Dear reviewer,

Previously, SEM images and EDS images were not realized at the same time.

So, we realized new SEM images which are matching now with EDS images. We changed Fig 7,8 and 10 according these new pictures.

Best regards,

Sébastien QUENARD